# Incidence, Clinical Features and Vaccination Coverage of Pneumococcal Disease in People Living with HIV: A Retrospective Cohort Study (2015–2024)

**DOI:** 10.3390/vaccines13121240

**Published:** 2025-12-13

**Authors:** Pere Medina, Elisa de Lazzari, Mateu Espasa, Lorena de la Mora, María Martínez-Rebollar, Ana González-Cordón, Montserrat Laguno, Alexy Inciarte, Juan Ambrosioni, Júlia Calvo, Alberto Foncillas, Abiu Sempere, Ivan Chivite, Leire Berrocal, Jose Luis Blanco, Esteban Martínez, José M. Miró, Josep Mallolas, Berta Torres

**Affiliations:** 1Department of Internal Medicine, Hospital Clínic, 170 Villarroel St, 08036 Barcelona, Spain; pemedina@clinic.cat; 2HIV Unit, Infectious Disease Service, Hospital Clínic Barcelona-Institut d’Investigacions Biomèdiques August Pi i Sunyer (IDIBAPS), University of Barcelona, 585 Gran Via de les Corts Catalanes, 08007 Barcelona, Spain; elazzari@recerca.clinic.cat (E.d.L.); rebollar@clinic.cat (M.M.-R.); mlaguno@clinic.cat (M.L.); ajinciar@clinic.cat (A.I.); ambrosioni@clinic.cat (J.A.); jcalvoj@clinic.cat (J.C.); foncillas@clinic.cat (A.F.); sempere@clinic.cat (A.S.); ichivite@clinic.cat (I.C.); berrocal@recerca.clinic.cat (L.B.); jlblanco@clinic.cat (J.L.B.); estebanm@clinic.cat (E.M.); jmmiro@clinic.cat (J.M.M.); mallolas@clinic.cat (J.M.); btorres@clinic.cat (B.T.); 3CIBER de Enfermedades Infecciosas (CIBERINFEC), Instituto de Salud Carlos III, 28029 Madrid, Spain; 4Department of Microbiology, Hospital Clínic, c/Villarroel 170, 08036 Barcelona, Spain; mespasa@clinic.cat; 5Reial Acadèmia de Medicina de Catalunya (RAMC), 08001 Barcelona, Spain

**Keywords:** pneumococcal disease, HIV, vaccination, incidence rate

## Abstract

**Introduction:** The incidence of pneumococcal disease (PD) in people living with HIV (PLWH) is higher than in the general population; therefore, this study aimed to analyze its incidence, clinical characteristics and vaccination coverage in PLWH. **Methods:** We conducted a retrospective, single-center study between 2015 and 2024 in Hospital Clínic, Barcelona, involving HIV patients who presented with PD during the study period (any patient with a microbiologically confirmed result). A descriptive analysis of cases was carried out and compared with patients who did not present PD during the study period. **Results:** A total of 177 episodes of PD were identified in 148 individuals, with a cumulative incidence of 1.7% (95% CI: 1.4–2.0). The median age at PD diagnosis was 45.9 (36–53) years; 64% of patients were Spanish-born; 50% of patients were men who have sex with men (MSM); the HIV transmission mode was intravenous drug use in 28% of cases; the median CD4 nadir was 181 (58–324) cells/mm^3^; the median CD4 prior to PD was 429 (240–663) cells/mm^3^; and the median peak HIV viral load (VL) was 176,839 (20,900–502,000) copies/mL. Intravenous drug use (OR 3.43; 95% CI 2.19–5.36; *p* < 0.001), peak HIV VL (OR 1.11; 95% CI 1.02–1.21; *p* = 0.011), and CD4 nadir (OR 0.92; 95% CI 0.87–0.98; *p* = 0.005) were independently associated with PD, and fifty-one percent of patients had not received any vaccination prior to their PD episode. **Conclusion:** PD incidence was high in our study and associated with poor immunological status. Research on new strategies to improve vaccination coverage and immunogenicity in PLWH is needed.

## 1. Introduction

Pneumococcal disease (PD) is an infection associated with high morbidity and mortality in the general population. In Spain, the incidence of pneumococcal pneumonia was 19.59 per 100,000 population for the period 2008–2017 [1], and the incidence of invasive pneumococcal disease was 9.91 cases per 100,000 inhabitants in 2023 [2].

Among people living with HIV (PLWH), the incidence of pneumococcal disease remains higher than in the general population, even in patients with a high CD4 count [3] and suppressed viral load [4]. However, some studies suggest that in virologically suppressed patients with CD4 counts > 350 cells/mm^3^, the clinical presentation and prognosis of pneumococcal pneumonia do not differ significantly from the general population [5].

In accordance with current Spanish vaccination guidelines, various vaccines are available against *Streptococcus pneumoniae*: a 23-valent polysaccharide vaccine (PPSV23) and three conjugate vaccines, covering 13 (PCV13), 15 (PCV15), and 20 serotypes (PCV20), respectively, which include most of those responsible for invasive pneumococcal disease (IPD) [6]. Although current international guidelines recommend systematic pneumococcal vaccination in all PLWH [7], the rate of immunization against *S. pneumoniae* in this population remains low [8,9].

In this study, we aimed to assess the incidence, clinical and immunovirological characteristics, severity, and vaccination coverage of pneumococcal disease among PLWH in a large single-center cohort over the last decade and to identify the factors associated with disease severity.

## 2. Methods

### 2.1. Study Design and Population

This is an observational, retrospective cohort study conducted at a single center, including individuals with HIV infection who presented with pneumococcal disease (PD) and were actively followed at the HIV Unit of Hospital Clínic of Barcelona at any time during the study period (January 2015 to June 2024). No additional exclusion criteria were applied to capture the full burden of pneumococcal infection within the cohort. A pneumococcal disease (PD) case was defined as any patient with a microbiologically confirmed result for *Streptococcus pneumoniae* via urinary antigen test or culture from any clinical sample obtained at Hospital Clínic. Results from samples collected at other centers were not available.

A control group was subsequently incorporated with the sole purpose of comparing epidemiological variables that remained unchanged over time and contextualizing the cases’ characteristics. Controls were defined as HIV patients who were in active follow-up at any time during the study period and who did not develop PD in the hospital setting during the specified period.

For cases with pneumococcal disease (PD), the inclusion date was defined as the date of their first PD episode. For controls, the inclusion date was defined as either the study start date (January 2015) or, for those enrolled later, the first visit before 1 July 2024.

Considering the difference in inclusion dates between cases and controls, only variables that remained unchanged over time were compared between the two groups. Due to the same rationale, vaccination coverage was described in the two groups but could not be compared between cases and controls. Vaccination was carefully revised in cases (meaning that if patients were vaccinated in a healthcare facility other than the hospital clinic, these data were also recorded), but in controls, data were restricted to hospital records.

Severe PD infection was defined based on whether ICU admission was required or if death occurred during PD episode hospitalization.

Invasive pneumococcal disease (IPD) was defined if patients had positive blood cultures for *S. pneumoniae* or other positive cultures in sterile fluids.

For vaccination coverage assessment, a complete pneumococcal vaccination schedule was defined according to national guidelines at study start time as either a combination of 13-valent conjugate vaccine (PCV13) and two doses of the 23-valent polysaccharide vaccine (PPSV23) separated by a minimum of five years (if PD occurred within five years from the administration of PCV13 and a single PPSV23, the scheme was also considered complete) or the administration of a single dose of the 20-valent conjugate vaccine (PCV20), according to the newly approved scheme [6]. If patients received any vaccine dose not fulfilling the complete scheme definition, vaccination was considered partial. Vaccine coverage percentages were reported prior to PD episode for cases, at the beginning of the study period for controls, and at the end of the study period for both cases and controls.

This manuscript was prepared following the STROBE (Strengthening the Reporting of Observational Studies in Epidemiology) guidelines for observational cohort studies.

### 2.2. Data Management and Statistical Analysis

To collect the data for this study, two different methods were used: specific data regarding streptococcus episodes were manually entered into an electronic Case Report Form (eCRF) in the REDCap system hosted at Hospital Clínic, and sociodemographic and clinical data were retrieved directly from Hospital Virtual (Hvirtual), the VIH Unit at Hospital Clinic’s electronic medical record system. Statistical analyses were performed using Stata version 18.

Qualitative variables were described using frequencies and percentages relative to the total number of valid values and compared between individuals presenting at least one episode and those without any using the Chi-squared or Fisher’s exact test. Quantitative characteristics were summarized using median and interquartile range (IQR) and compared between the two groups using the Wilcoxon rank sum test. Differential characteristics of severe episodes (vs. non-severe) were identified using Poisson regression models with clustered standard errors at the participant level. Unadjusted Risk Ratios (RRs) and their corresponding confidence intervals (Cis) were reported, with the episode incidence rate (IR) expressed as the number of episodes per 100,000 person-years. The cumulative incidence function of mortality, represented by the Kaplan–Meier failure estimate, was plotted along with the Cis. A logistic regression model adjusted for sex was used to identify the characteristics associated with the probability of hospital admission due to pneumococcal disease. All variables assessed in the simple models were entered in block into the multiple (final) model. Changes in the number of people with complete, partial, or no pneumococcal vaccination from the beginning to the end of the study were analyzed using the marginal homogeneity (Stuart–Maxwell) test and visualized using a Sankey plot. All tests were two-sided, and the confidence level was set at 95%.

## 3. Results

From January 2015 to June 2024, a total of 177 cases of *Streptococcus pneumoniae* infection were diagnosed in 148 PLWH among the 8803 individuals followed in our HIV cohort, yielding a cumulative incidence of 1.7% (95% CI: 1.4–2.0), equivalent to 1681 cases per 100,000 individuals. The incidence rate was 304.6 (95% CI: 259.63–349.57) cases per 100,000 person-years.

Of the 148 patients, 13 (9%) had two episodes and 4 (3%) had three or more episodes.

Among available microbiological tests for the first episode in each patient, 92/148 (62%) were diagnosed via positive culture, most frequently from respiratory samples (61/92, 66%) and blood cultures (26/92, 28%); a total of 73/148 (49%) had a positive urinary antigen for *S. pneumoniae*; and in 17/148 (11%), both the antigen test and culture were positive. For the total episodes observed during the study period, 29/177 (16%) were considered invasive pneumococcal disease resulting in a cumulative IPD incidence of 0.3% (95%CI: 0.2–0.5).

The baseline characteristics of PLWH who presented PD (cases) and the controls are shown in [Table 1].

Of the 148 patients who presented PD during the study period, 80% were assigned male at birth, though we observed a higher proportion of women among pneumococcal cases compared to controls (20% vs. 12%; *p* = 0.004). Median age when PD first occurred was 45.9 (36–53) years old and the median time from HIV diagnosis to the first episode was 12.4 years (2.5–21.5). A total of 64% of cases were Spanish-born, compared to 48% of controls (*p* < 0.001); 50% were men who have sex with men (MSM) and 45% were heterosexual compared to 75% and 22%, respectively, in the controls (*p* < 0.001). HIV transmission occurred via sexual contact in 70% of cases vs. 90% in controls (*p* < 0.001) and by intravenous drug use (IDU) in 28% vs. 9% in controls (*p* < 0.001).

Regarding immunovirological status, the median T CD4 lymphocyte nadir was 181 [58–324] cells/mm^3^, lower than in controls (322 [183–496]; *p* < 0.001), and the median peak viral load was 176,839 (20,900–502,000) copies/mL, higher than in controls (37,333 [450–189,900]; *p* < 0.001). When the first PD episode occurred, the median T CD4 lymphocyte count was 429 [240–663] cells/mm^3^ and only 65% of cases had an undetectable HIV viral load (<50 copies/mL). Of the 35% who had a detectable viral load at the time of PD presentation, ten patients (7%) were naïve to antiretroviral treatment.

Among the 148 cases, 16 (11%) had at least one additional cause of immunosuppression at the time of PD diagnosis and a total of 28 (19%) had a pre-existing lung disease, the most frequent being COPD (18/28, 64%). The presence of other immunosuppression causes and predisposing comorbidities for PD were not evaluated in the controls.

Risk factors associated with pneumococcal disease:

As some of the variables related to cases presented more frequently in women in our cohort (Appendix A), we created a logistic regression model adjusted for sex to control for potential confounding by female sex. The characteristics independently associated with pneumococcal disease likelihood were intravenous drug use as the acquisition mode (OR 3.43; 95% CI 2.19–5.36; *p* < 0.001), peak HIV viral load (OR 1.11; 95% CI 1.02–1.21; *p* = 0.011), and CD4 count nadir (OR 0.92; 95% CI 0.87–0.98; *p* = 0.005) [Table 2].

Severity of pneumococcal disease cases and comparison of characteristics [Table 3]:

A total of 26 out of 177 episodes (15%) were considered severe infections: 25 patients (14%) were transferred to the intensive care unit (ICU) during hospitalization, and 5 patients died during the episode, of whom 4 had been admitted to the ICU.

There were no differences in sex, age, or HIV transmission mode between severe and non-severe cases, though a higher proportion of Spanish-born individuals was observed to have severe episodes (88% vs. 63%; RR 3.50; 95% CI 1.08–11.29; *p* = 0.036). No significant differences were found in CD4 nadir, CD4 count at the time of pneumococcal infection, or proportion of patients with detectable viral load when PD occurred; however, a statistically higher CD4/CD8 ratio (medians: 0.6 vs. 0.4; RR 1.74; 95% CI 1.36–2.23; *p* < 0.001) was observed in severe patients. As expected, hospital stay was significantly longer in severe episodes: a median of 11 [7–35] days vs. 5 [1–9] days in non-severe cases (RR 1.03; 95% CI: 1.02–1.03; *p* < 0.001).

There were no differences in the proportion of PD diagnosed via urinary antigen, in positive cultures, or in cases considered IPD. There were also no differences in vaccination coverage between groups.

Mortality in pneumococcal disease cases:

Mortality was assessed at 30 days, 3 months, and 1 year following the index episode, regardless of hospitalization status. Eleven patients (7%) died within the first year: three in the first month, three between months 1 and 3, and five between months 3 and 12. The cumulative mortality incidence was 2% (95% CI: 0.7–6.2) at one month, 4% (95% CI: 1.9–8.9) at three months, and 8% (95% CI: 4.3–13.5) at one year.

Vaccination coverage [Figure 1]:

Despite vaccination recommendations, pneumococcal vaccination coverage remained suboptimal. Regarding vaccination status prior to the first PD episode, only 20% (29/148) of patients had completed the full schedule, with the majority receiving the PCV13 + PPSV23 combination (28/29; 97%). A total of 43 out of 148 (29%) had received partial vaccination and 76/148 (51%) had not received any dose when PD first occurred.

The median time from the last vacc”ne d’se to the first pneumococcal episode was 41.7 [10.9–87.6] months.

In the control group, only 47/8499 (1%) had a registered complete vaccination course at the beginning of the study period, 2288/8499 (27%) had a partial course, and 6164/8499 (73%) did not have any pneumococcal vaccine registered in the hospital’s clinical records.

At the end of the study period end, vaccination coverage significantly increased in both cases and controls, reaching complete vaccination in 51/148 (34%) of cases (12% had received a dose of PCV20) and in 1936/8499 (23%) of controls (7% being PCV20).

## 4. Discussion

In this study, we describe the incidence and features of a well-characterized cohort of HIV patients presenting with an episode of pneumococcal disease diagnosed in the hospital setting in the last decade.

In our cohort of PLWH, we observed a markedly higher incidence of PD compared to the general population. In a study conducted in the region of Catalonia, Spain, the pneumococcal pneumonia incidence in people over 50 years old was 90.7 cases per 100,000 person-years for the period 2017–2018 [10]. In the same study, the incidence for HIV patients >50 years old was 423.7 cases per 100,000 person-years, higher than the incidence observed in our study. The median age at PD presentation in our cohort was 49 years old, highlighting that half of the patients were infected before the age of 50, much younger that the mean age of 63.4 years reported for PD in the general population in Spain [11]. The incidence in our study is lower than the 1529 community-acquired pneumonia (CAP) cases per 100,000 person-years follow-up reported in a study of a Dutch HIV cohort conducted between 2008 and 2017 [3]. In that study, all episodes of CAP registered in the electronic patient file were included, in comparison to our study, where only patients who presented to the hospital were included.

A similar, although higher, incidence to our study was observed in a recent Danish study which included patients from 2015 to 2021 and reported a pneumococcal infection rate of 5.5 cases per 1000 person-years. In this last study, data were collected from a national database, and only virologically suppressed HIV patients were included [9].

The well-described risk factors for pneumococcal disease in HIV patients, like low CD4 nadir, not taking antiretroviral therapy, or intravenous drug use being the HIV acquisition mode [12,13], were also seen in the included patients’ baseline characteristics, demonstrating that the incidence of pneumococcal disease in HIV patients in the last decade remains high as a consequence of suboptimal infection control; 35% of patients had a detectable viral load at the time of PD in our study, despite being in the universal antiretroviral treatment era, and had poor HIV control in the past, evidenced by a lower CD4 nadir in cases in contrast to the controls in our study.

When we compared the epidemiological and HIV-related characteristics of the patients who were diagnosed with pneumococcal disease versus those who were not, we observed a higher proportion of female sex in PD cases.

As explained in the Results, women in our cohort currently receiving follow-up are reported to be older, to have a longer time since HIV diagnosis, to have a higher frequency of intravenous drug use as the mode of acquisition, and to experience more virological failure than men [14]. For that reason, we performed a multiple logistic regression model adjusted by sex to eliminate sex as a factor of confusion. In this model, intravenous drug use as an acquisition mode, lower CD4 nadir, and higher HIV peak VL were still associated with PD cases.

Acquiring HIV by past intravenous drug use is not necessarily related to active drug use; however, this population often experiences more socioeconomic difficulties, like unstable housing or unemployment, which contribute to lower treatment adherence and care linkage and poorer overall health status [15]. Additionally, a low CD4 nadir in HIV infection has been independently associated with significant B cell impairment that is not fully restored with antiretroviral treatment (reviewed in [16]), predisposing patients to more invasive bacterial infections than the general population [17]. Additionally, high peak viral load could be an indicator of higher immune activation and T and B cell exhaustion, which could also increase susceptibility to non-AIDS events, including bacterial infections [18].

Furthermore, a substantial proportion of cases were classified as severe: 14% required ICU admission compared to the estimated 12.9% ICU admission rate among the general population with invasive pneumococcal disease [19]. Considering that not all of our cases were classified as invasive, this finding points to a potentially greater clinical impact of PD among PLWH. However, the 30-day mortality rate observed in our cohort (2%) was slightly lower than that in the general population under 50 years of age requiring hospitalization for pneumococcal disease in Spain (3–4%) [20]. This finding is in line with that of a study conducted in the same HIV cohort as our center, where pneumococcal disease prognosis in suppressed patients with >350 cells/mm^3^ was similar to HIV-negative patients [5].

No significant differences were observed between severe and non-severe cases, except that there was a higher percentage of Spanish-born patients and that the CD4/CD8 ratio was higher in severe patients. Although this last finding was unexpected, it could be explained by the CD8 T cell depletion observed in severe bacterial infections, rather than by a decrease in CD4 counts. In fact, CD8 depletion has been associated with poor prognosis in sepsis [21]. In our cohort, CD4 counts were consistently similar between severe and non-severe cases, whereas the former showed lower absolute CD8 counts, supporting this hypothesis.

We did not find any difference in vaccination coverage between severe and non-severe cases either.

Use of the 13-valent pneumococcal conjugate vaccine was approved in subjects over 18 years old by the European Medicines Agency in July 2013 due to its better immunogenic response than PPSV23 alone; however, at the present moment, studies exploring its clinical efficacy in PLWH are scarce.

Introducing sequential pneumococcal vaccination with PCV13 in children and then in the adult population has been shown to reduce IPD risk in PLWH in some retrospective studies [22]; however, studies evaluating the direct effect of sequential vaccination of PCV13 followed by PPSV23 in the HIV population are scarce and have conflicting results [23]. As a matter of fact, immunogenicity and clinical protection of vaccines in PLWH are suboptimal compared to the general population, and less than a quarter of patients with HIV exhibit protection at month 12 after vaccination with PCV13 followed by PPSV23 [24].

Local recommendations about pneumococcal vaccination [6] from 2017 in the Catalan region reported a PPSV23 coverage of 53% in people <65 years with at least one risk factor, but the coverage of PCV13 in immunocompromised patients was only 3% [23]. As explained in the Methodology section, we defined a complete vaccination scheme as receiving the combination of PCV13 and PPS23V. The low complete scheme coverage observed in our study at baseline could be because PCV13 was approved in Europe in July 2013 and the study start date was January 2015; notwithstanding this, 51% of cases had not received any pneumococcal vaccination before the PD episode. Moreover, almost half of the patients who presented with PD had previously received any vaccine and the median time from last vaccine to PD episode was more than three years, which could indicate the low immunogenicity that HIV patients present with after vaccination, especially with a CD4 count nadir <200 cells/mm^3^ [25] or when vaccination is received with low CD4 counts or with detectable viral load [26].

Data on vaccination coverage in controls should be interpreted cautiously, as only the recorded vaccines received in the hospital setting were considered; vaccination coverage at the end of the study period could give a better and more realistic approach. Finally, for both cases and controls, vaccination percentage at the end of the study significantly increased, with some patients being vaccinated with PCV20, approved in Europe in March 2022. However, almost a third of patients who presented PD remained completely unvaccinated at the end of study period.

This study had several limitations: First, due to the retrospective nature of the study, some patients may have been misclassified as controls despite having experienced pneumococcal infection before the study period. Second, since the control group was not included in the initial study design, it was not possible to compare all of the variables collected in the cases. Additionally, vaccination coverage in controls may have been underestimated, as data were limited to hospital records. Third, only diagnoses in the hospital setting were collected, so incidence in our study was limited to pneumococcal disease in this specific context. Finally, a key limitation was the lack of comparison with the general population, as our analyses were restricted to comparisons within the HIV cohort.

The strengths of this study include its large sample size, as Hospital Clínic hosts one of the largest HIV cohorts in Europe, with 6600 patients in active follow-up. In addition, we provide updated evidence in a contemporary population of people living with HIV, thereby reflecting current clinical and epidemiological conditions. Importantly, we also offer longitudinal data on pneumococcal vaccine coverage across a well-defined HIV cohort over a 10-year period, contributing valuable insights into vaccination practices and their evolution in long-term care.

Moreover, microbiological results were retrieved automatically, minimizing errors in patient inclusion. Finally, the single-center design enabled thorough chart review and data verification, further reducing the documentation error likelihood.

## 5. Conclusions

In conclusion, despite the widespread introduction of universal antiretroviral treatment, pneumococcal disease has remained highly prevalent over the last decade among PLWH, particularly in those with poor immunological status and inadequate virological control. There is an urgent need to strengthen systematic pneumococcal vaccination efforts in this population. Furthermore, prospective studies are required to evaluate the impact of new conjugate vaccines, such as PCV20, in people living with HIV, as well as to explore alternative strategies that may offer greater immunogenicity.

## Figures and Tables

**Figure 1 vaccines-13-01240-f001:**
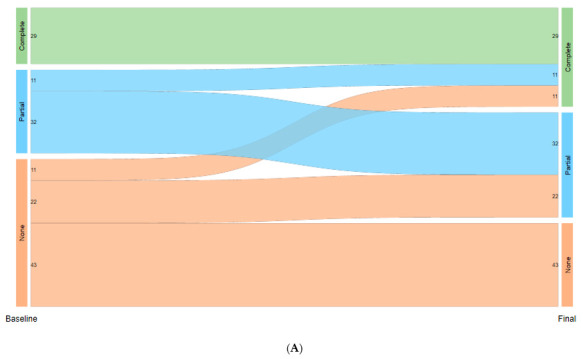
Sankey plot of vaccination coverage in cases (**A**) and controls (**B**). (**A**) Vaccination in cases at baseline (PD episode) and at end of study period (June 2024). (**B**) Vaccination in controls at baseline (study start date or first visit during study period) and at end of study period (June 2024). The green bars represent individuals with complete vaccination, the orange bars represent those with partial vaccination, and the blue bars represent those not vaccinated. The flow between bars indicates transitions in vaccination status over time.

**Table 1 vaccines-13-01240-t001:** Baseline sociodemographic and viro-immunological characteristics. Note that for cases with pneumococcal disease (PD), the inclusion date was defined as the date of their first PD episode. For controls, the inclusion date was defined as either the study start date (January 2015) or, for those enrolled later, the first visit before 1 July 2024. Considering the difference in inclusion dates between cases and controls, only variables that remained unchanged over time were compared between the two groups. Due to the same rationale, vaccination coverage was described in the two groups but could not be compared between cases and controls. MSM: men who have sex with men; IDU: intravenous drug use; ART: antiretroviral treatment; PD: pneumococcal disease; IS: immunosuppression.

Variable	Cases (*n* = 148)	Controls (*n* = 8499)	*p*-Value
**Sex at birth,** * ** n** * **(%)**			**0.004**
Male	118 (80)	7446 (88)	
Female	30 (20)	1053 (12)	
**Age at first episode,** **median (IQR)**	45.9 (36.1–53.3)	--	
**Birthplace,** * ** n** * **(%)**			**<0.001**
Spain	89 (64)	3865 (48)	
Other	50 (36)	4172 (52)	
**Sexual orientation,** * ** n** * **(%)**			**<0.001**
MSM	55 (50)	5550 (73)	
Heterosexual	50 (45)	1643 (22)	
Bisexual	6 (5)	438 (6)	
**Transmission mode,** * ** n** * **(%)**			**<0.001**
Sexual	101 (70)	7570 (90)	
IDU	41 (28)	735 (9)	
**Transfusion**	0 (0)	32 (<1)	
**Vertical**	3 (2)	31 (<1)	
**CD4 nadir, median (IQR)**	181 (58–324)	322 (183–496)	**<0.001**
**HIV VL peak (cp/mL), median (IQR)**	176,839 (20,900–502,000)	37,333 (450–189,900)	**<0.001**
**Years between HIV+ and PD, median (IQR)**	12.4 (2.5–21.5)	--	
**Years between ART start and PD, median (IQR)**	4.8 (0–15.8)	--	
**Naïve to ART at PD,** * ** n** * **(%)**	11 (7)	--	
**CD4 at PD, median (IQR)**	429 (239–662)	--	
**CD8 at PD, median (IQR)**	844 (602–1263)	--	
**CD4/CD8 at PD,** **median (IQR)**	0.5 (0.3–0.7)	--	
**Suppressed HIV VL at PD,** * ** n** * **(%)**			
No	45 (35)	--	
Yes	84 (65)	--	
**Additional cause of IS present,** * ** n** * **(%)**	16 (11)	--	
**Pre-existing lung disease,** * ** n** * **(%)**	21 (14)	--	
**Vaccination coverage,** * ** n** * **(%)**			
None	76 (51)	6164 (73)	
Partial	43 (29)	2288 (27)	
Complete	29 (20)	47 (1)	

**Table 2 vaccines-13-01240-t002:** Characteristics associated with presence of pneumococcal disease. Simple and multiple (final) logistic regression models adjusted for sex. VL: viral load; IDU: intravenous drug use.

Variable	OR (95% CI)—Simple Model	*p*-Value	OR (95% CI)—Multiple Model	*p*-Value
**Birthplace**		<0.001		0.438
Other	2.06 (1.38–3.06)		1.19 (0.77–1.82)	
Spain	1		1	
**HIV transmission mode**		<0.001		**<0.001**
Other	4.58 (3.01–6.95)		3.43 (2.19–5.36)	
IDU	1		1	
**CD4 nadir ^1^**	0.87 (0.82–0.91)	<0.001	0.92 (0.87–0.98)	**0.005**
**HIV VL peak ^2^**	1.20 (1.11–1.29)	<0.001	1.11 (1.02–1.21)	**0.011**

^1^: Odds Ratio per 50-unit increase; ^2^: Odds Ratio per 0.5-unit increase; number of observations: 7928.

**Table 3 vaccines-13-01240-t003:** Comparison between severe and non-severe cases. Severe pneumococcal disease was determined based on whether intensive care unit admission was required or if death occurred during hospitalization for the pneumococcal disease episode. Risk Ratios were calculated using appropriate statistical models. MSM: men who have sex with men; IDU: intravenous drug use; ART: antiretroviral treatment; PD: pneumococcal disease; IS: immunosuppression.

Variable	Severe (*N* = 26)	Non-Severe (*N* = 151)	Risk Ratio (95% CI)	*p*-Value
**Sex at birth,** * ** n** * **(%)**				0.525
Male	21 (81)	114 (75)	1	
Female	5 (19)	37 (25)	0.77 (0.34–1.75)	
**Age at first episode** **median (IQR)**	52 (44–58)	49 (39–58)	1.01 (0.99–1.04)	0.347
**Birthplace,** * ** n** * **(%)**				0.036
Spain	21 (88)	87 (63)	1	
Other	3 (12)	51 (37)	3.50 (1.08–11.29)	
**HIV transmission mode,** * ** n** * **(%)**				0.615
Sexual	14 (56)	98 (66)	1	
IDU	10 (40)	46 (31)	1.43 (0.70–2.93)	
Vertical	1 (4)	5 (3)	1.33 (0.35–5.04)	
**CD nadir** **median (IQR)**	139 (49–272)	165 (35–301)	1 (1.00–1.00)	0.839
**CD4 at PD** **median (IQR)**	337 (175–654)	325 (160–586)	1 (1.00–1.00)	0.708
**CD8 at PD** **median (IQR)**	630 (355–1168)	822 (480–1273)	1 (1.00–1.00)	0.871
**CD4/CD8 at PD** **median (IQR)**	0.6 (0.1–1)	0.4 (0.2–0.7)	1.74 (1.36–2.23)	**<0.001**
**Suppressed HIV VL at PD,** * ** n** * **(%)**				0.452
No	12 (50)	58 (42)	1	
Yes	12 (50)	81 (58)	0.75 (0.36–1.58)	
**Additional cause of IS,** * ** n** * **(%)**	3 (12)	16 (11)	1.08 (0.35–3.39)	0.889
**Invasive pneumococcal disease,** * ** n** * **(%)**	6 (23)	23 (15)	1.53 (0.67–3.50)	0.313
**Hospital stay (days), median (IQR)**	11 (7–35)	5 (1–9)	1.03 (1.02–1.03)	**<0.001**
**Vaccination coverage,** * ** n** * **(%)**				0.190
None	14 (54)	71 (47)	1	
Partial	11 (42)	47 (31)	1.15	
Complete	1 (4)	33 (22)	0.18 (0.02–1.32)	

## Data Availability

The datasets generated during the current study are available from the corresponding author on reasonable request. Access to the data will be provided if considered justified by the journal’s editors or reviewers.

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
