# Peer review of "Incidence, Clinical Features and Vaccination Coverage of Pneumococcal Disease in People Living with HIV: A Retrospective Cohort Study (2015–2024)"

_vaccines, 2025, doi:10.3390/vaccines13121240_

Round 1

Reviewer 1 Report

Comments and Suggestions for Authors

I commend the authors for addressing a clinically and epidemiologically relevant topic. However, the manuscript requires several methodological refinements and improvements in its overall presentation.

The introduction is well contextualized, but the scientific gap that justifies the study should be defined more explicitly. It would benefit from a clearer explanation of what remains unknown, particularly regarding the behavior of pneumococcal disease after the introduction of higher-valency conjugate vaccines and the relationship between CD4 nadir, vaccine response, and residual risk. The study objectives should also be stated more clearly and directly.

The Methods section requires substantial revision. The authors should explicitly state that the study followed STROBE guidelines, which are the international standard for observational cohort studies. Stating this clearly would strengthen methodological transparency. Moreover, the study design should be identified more assertively at the beginning of the Methods section. Although the manuscript mentions that the study is retrospective and single-center, it is not made explicit that it is an observational retrospective cohort study. This information should appear upfront, and the title should ideally reflect the study design, as recommended by STROBE, by indicating that the work is a retrospective cohort study.

The selection of controls needs clearer detail. Although controls were defined as patients followed during the study period who did not develop pneumococcal disease, the criteria for identifying this group are not fully transparent. The authors should specify the inclusion and exclusion criteria applied to controls and provide justification for not using matching. Using any patient without pneumococcal disease as a control, without matching or adjusting for time-dependent covariates, may introduce selection bias. A participant flow diagram should also be included to show how cases and controls were identified, selected, and retained in the final analytic sample. In addition, the manuscript does not report how missing data were handled, which is essential in a retrospective study of this size.

The presentation of the Results requires improvement. Tables display formatting inconsistencies, misalignment, inconsistent percentage notation, and captions that do not adequately describe the content. Standardizing table formatting, providing clearer headers, and adding more descriptive legends would improve clarity and facilitate interpretation.

The Discussion could be strengthened by explicitly highlighting the novel contribution of this study, since several findings echo results reported in other cohorts. The unexpected observation of a higher CD4/CD8 ratio in severe cases also deserves a more nuanced interpretation supported by literature or a clearer immunological rationale. The limitations section is generally appropriate but should acknowledge the likely underestimation of vaccination coverage among controls due to the lack of external vaccination records, as well as the inability to identify episodes diagnosed outside the hospital setting.

The conclusion is appropriate but overly brief. It would be strengthened by stating more explicitly the practical implications for pneumococcal vaccination strategies in people living with HIV and reinforcing the need for improved immunization efforts in this population. Emphasizing the importance of future research on the clinical effectiveness and immunogenicity of PCV20 in PLWH would also align the conclusion with emerging research priorities.

Finally, the reference list needs revision. Several references are incomplete, not formatted according to the journal’s style, and do not include active URLs where required.

Reviewer 2 Report

Comments and Suggestions for Authors

The manuscript is devoted to the important public health problem – the substantial higher morbidity of pneumococcal disease (PD) among HIV positives. It is important not only because the HIV-infected people are suffering but because they are sources of this infection for other people for a prolonged time.

Shortages of the manuscript:

  1. The producers of all mentioned vaccines and all the diagnostic kits and exact types of these preparations are not shown in the manuscript. But it is very important because significant part of vaccinated persons became ill.
  2. The age distribution of the PD patients is not shown in the manuscript. But it may be significant.
  3. In the line 259 there is an expression “female sex”, and it is not clear what does it mean in this case.
  4. In the lines 271-272 it is stated that 14% is slightly higher than 12,9% and this very small difference which is not supported statistically, is further seriously discussed.
  5. The different fonts are used between lines 293 and 300.
  6. The last paragraph contains some useful proposals about vaccination strategies but did not present any conclusions and even suggestions about strong necessity to use the most effective pneumococcal vaccines for HIV positives and recommendations about about the obligate vaccination of these persons because in other case they will continue to increase the dissemination of this pathogen among the whole population.                                                                                             The manuscript should be corrected according to these comments
Comments on the Quality of English Language

The manuscript is devoted to the important public health problem – the substantial higher morbidity of pneumococcal disease (PD) among HIV positives. It is important not only because the HIV-infected people are suffering but because they are sources of this infection for other people for a prolonged time.

Shortages of the manuscript:

  1. The producers of all mentioned vaccines and all the diagnostic kits and exact types of these preparations are not shown in the manuscript. But it is very important because significant part of vaccinated persons became ill.
  2. The age distribution of the PD patients is not shown in the manuscript. But it may be significant.
  3. In the line 259 there is an expression “female sex”, and it is not clear what does it mean in this case.
  4. In the lines 271-272 it is stated that 14% is slightly higher than 12,9% and this very small difference which is not supported statistically, is further seriously discussed.
  5. The different fonts are used between lines 293 and 300.
  6. The last paragraph contains some useful proposals about vaccination strategies but did not present any conclusions and even suggestions about strong necessity to use the most effective pneumococcal vaccines for HIV positives and recommendations about about the obligate vaccination of these persons because in other case they will continue to increase the dissemination of this pathogen among the whole population.

The manuscript should be corrected according to these comments

Reviewer 3 Report

Comments and Suggestions for Authors

The authors investigated the incidence, severity, and vaccination coverage of pneumococcal disease among individuals living with HIV and demonstrated that intravenous drug use, peak of HIN viral load, and CD4 nadir were independently associated with PD. However, no significant association was found between vaccination coverage and either the incidence or severity of PD like due to low vaccination rates. The results are of interest, although certain aspects require further clarification.

  1. Please ensure consistency in the number of decimal places used for percentages.
  2. Streptococcus pneumoniae is spelled out in full upon first mention followed by the abbreviated form S. pneumoniae in subsequent references.
  3. Line 146, 182[59-324] does not consistent with the data presented in table 1.
  4. The vaccination coverage presented in table 1 should be compared by statistical analysis.
  5. Lines 331, with 6,600 patients in active follow-up, is this data correct?
